# Gene expression phylogenies and ancestral transcriptome reconstruction resolves major transitions in the origins of pregnancy

Katelyn Mika[1,2], Camilla M Whittington[3], Bronwyn M McAllan[4], Vincent J Lynch[5]*

[1]Department of Human Genetics, University of Chicago, Chicago, United States; [2]Department of Organismal Biology and Anatomy, University of Chicago, Chicago, United States; [3]School of Life and Environmental Sciences, University of Sydney, Sydney, Australia; [4]Charles Perkins Centre,University of Sydney, Sydney, Australia; [5]Department of Biological Sciences, University at Buffalo, State University of New York, Buffalo,Newyork, United States

**\*For correspondence:**
vjlynch@buffalo.edu

**Competing interest:** The authors declare that no competing interests exist.

**Abstract** Structural and physiological changes in the female reproductive system underlie the origins of pregnancy in multiple vertebrate lineages. In mammals, the glandular portion of the lower reproductive tract has transformed into a structure specialized for supporting fetal development. These specializations range from relatively simple maternal nutrient provisioning in egg-laying monotremes to an elaborate suite of traits that support intimate maternal-fetal interactions in Eutherians. Among these traits are the maternal decidua and fetal component of the placenta, but there is considerable uncertainty about how these structures evolved. Previously, we showed that changes in uterine gene expression contributes to several evolutionary innovations during the origins of pregnancy (Mika et al., 2021b). Here, we reconstruct the evolution of entire transcriptomes ('ancestral transcriptome reconstruction') and show that maternal gene expression profiles are correlated with degree of placental invasion. These results indicate that an epitheliochorial-like placenta evolved early in the mammalian stem-lineage and that the ancestor of Eutherians had a hemochorial placenta, and suggest maternal control of placental invasiveness. These data resolve major transitions in the evolution of pregnancy and indicate that ancestral transcriptome reconstruction can be used to study the function of ancestral cell, tissue, and organ systems.

## Editor's evaluation

Mika and colleagues reconstruct the evolution of uterine endometrial transcriptomes during pregnancy from 23 diverse species of mammals that differ with respect to their degree of placental invasiveness. Through this analysis, the authors infer that the eutherian mammal ancestor had an invasive mode of placentation and that the degree of invasiveness of placentation is reflected on uterine endometrial gene expression during pregnancy. Thus, phylogenetic analysis of gene expression profiles of different mammals groups them on the basis of the degree of placental invasiveness, a quite striking finding.

## Introduction

Studies of the fossil record have revealed in fine detail major stages in the origin and diversification of evolutionary novelties (structures) and innovations (functions) such as the vertebrate limb and skull

**Figure 1.** Evolution and arrangement of the maternal-fetal interface and degree of placental invasion in viviparous Amniotes. (**A**) Phylogenetic relationships among amniote lineages, mammalian lineages labeled. Major evolutionary steps in the evolution of pregnancy are shown for Mammalia, Theria, and Eutheria. (**B**) Placenta classification based on the arraignment of the maternal-fetal interface and degree of invasiveness. Epitheliochorial placenta, in which placental invasion does not occur and the barrier between maternal blood and the chorion (tan cells) consists of maternal vascular endothelium (red cells) and uterine epithelium (blue cells). Endotheliochorial placenta, in which the placenta invades through the uterine epithelium and the barrier between maternal blood and the chorion consists of the maternal vascular endothelium. Hemochorial placenta, in which the placenta invades through both the uterine epithelium and vascular endothelium, and maternal blood directly bathes the chorionic villi. Examples of species with each type of placenta are also shown. While the egg passes through the glandular uterine portion of the oviduct in oviparous species, the immunologically inert shell (black) prevents direct contact between maternal and fetal tissues.

(*Abzhanov, 2015*; *Hirasawa and Kuratani, 2015*), the turtle shell (*Lyson and Bever, 2020*), feathers (*Chen et al., 2015*), and flowers (*Chanderbali et al., 2016*). Many features of soft tissues and macromolecular structures, however, are lost during the fossilization process and thus leave little to no trace in the fossil record. To reconstruct the history of these characters, including DNA and amino acid sequences, morphology, physiology, and behavior, among others, evolutionary studies have traditionally relied on comparative (phylogenetic) methods such as parsimony or model-based maximum likelihood or Bayesian inference (*Lewis and Olmstead, 2001*). These methods infer ancestral characters from the distribution of character states among extant species, the latter of which also allows increased model complexity including unequal character state transitions (*Lewis and Olmstead, 2001*; *Pagel and Cunningham, 1999*), variable rates among sites and branches (*Galtier, 2001*; *Yang, 1994*; *Yang, 1993*), and character-dependent diversification (*Maddison et al., 2007*).

Extant mammals span crucial transitions in the origins of pregnancy (*Figure 1A*) and are an excellent system in which to explore the origins of evolutionary novelties. The platypus and echidna (monotremes), for example, are oviparous (egg-laying) but retain the egg in the glandular portion of uterus for about two weeks. During this period, the developing embryo is nourished by uterine secretions (matrotrophy) delivered through a simple yolk sac placenta (*Hughes and Hall, 1998*; *Renfree and Shaw, 2013*). Viviparity (live-birth) evolved in the stem-lineage of therian mammals, but marsupials and eutherians have very different reproductive strategies, particularly in the ontogenetic origins of the definitive placenta and the arrangement of the maternal-fetal interface (*Freyer and Renfree,*

**Table 1.** Inferences of placental invasiveness in the eutherian ancestor 1880–2021.
Note that we follow *Mossman, 1991*, p. 156 and define a placenta as 'any intimate apposition or fusion of the fetal organs to the maternal (or paternal) tissues for physiological exchange'.

| Study | Data | Method(s) | State |
|---|---|---|---|
| *Turner, 1876* | – | Ontogeny | Epitheliochorial |
| *Haeckel, 1883* | – | Ontogeny | "Non-Invasive" |
| *Minot, 1891* | | Ontogeny | Hemochorial |
| *Wislocki, 1929* | – | Implicit Parsimony? | Hemochorial |
| *Hill, 1997* | – | Ontogeny | Epitheliochorial |
| *Mossman, 1991* | | Ontogeny | Endotheliochorial |
| *Portmann, 1938* | – | Ontogeny | Hemochorial |
| *Martin, 1969* | – | Ontogeny / Implicit Parsimony | Endotheliochorial |
| *Kihlström, 1972* | – | Gestation length / Placenta-type Correlation | (Endothelio/hemo)chorial |
| *Luckett, 1976*; *Luckett, 1975*; *Luckett, 1974* | – | Ontogeny | Epitheliochorial |
| | 14 taxa | Ontogeny / Implicit Parsimony | Endotheliochorial |
| *Vogel, 2005* | 22 taxa | Implicit Parsimony | Hemochorial |
| *Elliot and Crespi, 2006* | 88 taxa | Maximum Likelihood | Hemochorial |
| *Mess and Carter, 2006* | 36 taxa | Maximum Parsimony | (Endothelio/hemo)chorial |
| *Wildman et al., 2006* | 44 taxa | Maximum Parsimony & Maximum Likelihood | Hemochorial |
| *Martin, 2008* | 18 taxa | Parsimony | Endotheliochorial (ordered) or (Endothelio/hemo)chorial (unordered) |
| *Elliot and Crespi, 2009* | 334 taxa | Maximum Parsimony & Maximum Likelihood | (Endothelio/hemo)chorial (MP) or Hemochorial (ML) |
| *McGowen et al., 2014* | 66 taxa | Maximum Likelihood | (Superficial) Hemochorial |

*2009*; *Renfree and Shaw, 2013*; *Renfree, 2010*). In most marsupials, the embryonic portion of the placenta is derived from the yolk sac, which may come into direct contact with but does not invade the maternal endometrium. While the yolk sac has essential functions during early pregnancy in eutherians, the definitive placenta is derived from chorion and allantois (chorioallantois) and varies in its degree of endometrial invasion (*Swanson and Skinner, 2018*; *Figure 1B*). Thus, matrotrophy and a yolk-sac derived placenta were present in the mammalian stem-lineage and transitioned to a chorioallantoic placenta in eutherians.

Unfortunately, most characters related to pregnancy and the nature of the maternal-fetal interface leave little to no trace in the fossil record. Thus, studies exploring the evolution of pregnancy have relied on comparing morphological differences between extant mammals to reconstruct the steps in the origins of pregnancy. These comparative analyses have used multiple methods to reconstruct the arrangement of the mammalian maternal-fetal interface and reached contradictory conclusions about the degree of placental invasiveness in the eutherian ancestor, a debate which has persisted since at least 1876 (*Table 1*). Here, we use comparative transcriptomics and maximum likelihood to infer ancestral gene expression states (ancestral transcriptome reconstruction) from 23 amniotes with different parity modes and degrees of placental invasion to identify the evolutionary history of the maternal-fetal interface. We found strong evidence that the last common ancestor of eutherian mammals had an invasive hemochorial placenta, as well as convergence in gene expression profiles during the independent evolution of non-invasive epitheliochorial placentas in marsupials and some eutherian lineages. These data indicate that the degree of placental invasion can be inferred from

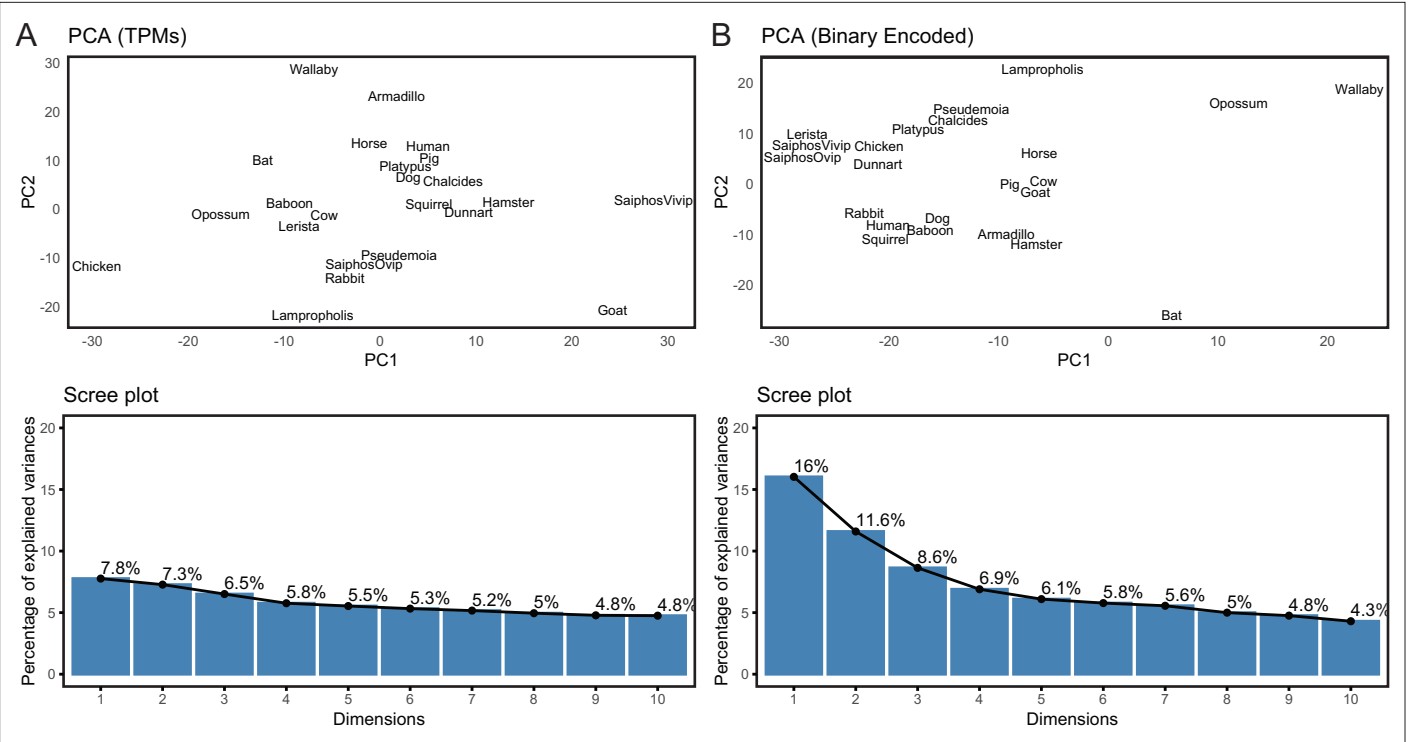

**Figure 2.** Binary encoding uncovers phylogenetic signal in transcriptome data. (**A**) Upper, Principal Component Analysis (PCA) of gene expression levels (TPMs) grouped species randomly, consistent with significant noise. Lower, scree plot showing percent of variance explained by dimensions 1–10. Note that the flat curve of variance explained indicates there are no significant dimensions to the PCA. (**B**) Upper, logistic PCA of the binary encoded endometrial transcriptome dataset groups species by phylogenetic relatedness, indicating significant noise reduction in the binary encoded dataset. Lower, screen plot showing percent of variance explained by dimensions 1–10. Note that the 'elbow' of the graph is around dimension 4, suggesting that dimensions 1–4 of the PCA are significant.

The online version of this article includes the following source data for figure 2:

**Source data 1.** file 1.

endometrial gene expression profiles and suggest that placental invasiveness is regulated by gene expression profiles in the maternal endometrium rather than the fetal portion of the placenta.

# Results

## Endometrial gene expression profiling

We previously assembled a collection of transcriptomes from the pregnant or gravid uterine endometrium of therian mammals with varying degrees of placental invasiveness, as well as a monotreme (platypus), a bird, as well as viviparous, oviparous, and reproductively bi-modal lizards (*Marinić et al., 2021*). The complete dataset includes expression information for 21,750 genes from 23 species (*Figure 2—source data 1*). Principal Component Analysis (PCA) based on gene expression levels in transcripts per million (TPMs), using a variant of PCA that extends PCA to binary data, generally grouped species randomly (*Figure 2A*), consistent with noise in gene expression data overwhelming phylogenetic signal. Therefore, we transformed quantitative gene expression values in transcripts per million (TPM) into discrete character states – Genes with TPM ≥2.0 were coded as expressed (state = 1), genes with TPM <2.0 were coded as not expressed (state = 0), and genes without data in specific species coded as missing (?) (*Marinić et al., 2021*; *Mika et al., 2021a*). In contrast to PCA based on TPM values, PCA of the binary encoded endometrial transcriptome dataset grouped species by phylogenetic relatedness (*Figure 2B*), indicating significant noise reduction in the binary encoded dataset.

## Phylogenetic analyses of endometrial transcriptomes

We used IQ-TREE to infer the best fitting model of character evolution and the maximum-likelihood (ML) phylogeny. The best-fit model of character evolution was a general time reversible model for binary data (GTR2) with character state frequencies optimized by maximum-likelihood (FO) and rate heterogeneity across sites accommodated with the FreeRate model that had three rate categories (R3). Next, we assessed the tree topology and branch support metrics for the ML phylogeny inferred using the binary encoded endometrial gene expression dataset and the GTR2 +FO + R3 model. The ML phylogeny (*Figure 3A* and ) generally followed taxonomic relationships (*Figure 3B*); however, four particularly discordant relationships within therian mammals were inferred with high support: (1) Rather than grouping marsupials into a monophyletic sister-clade to the eutherians, opossum and wallaby were placed as sister-species within the Boreoeutheria; (2) Armadillo groups within the Euarchontoglires rather than as sister to all other eutherians; (3) Bat groups within the Euarchontoglires rather than within Laurasiatheria; and (4) Dog groups within the Euarchontoglires rather than with Laurasiatheria (*Figure 3A/B*). We also used multiple non-parametric topology tests to directly compare the ML tree to alternative trees with the correct phylogenetic placement of these four discordant lineages, all of which rejected alternative 'corrected' trees in favor of the (*Figure 3C*). Remarkably while these discordant relationships are incorrect with respect to the species phylogeny, they are correct with respect to placenta-type, that is, species form well-supported clades based on their degree of placental invasiveness: Wallaby and opossum, which have epitheliochorial placentas similar to ungulates forms a clade with ungulates, Armadillo, which has an invasive placenta (*Carter and Enders, 2004*; *Chavan and Wagner, 2016*), forms a clade with the other species with hemochorial placentas, and dog, which has an invasive endotheliochorial placenta, forms a clade with Euarchontoglires that have invasive hemochorial placentas.

## Ancestral transcriptome reconstruction and fuzzy C-Means clustering

We also used IQ-TREE and the species phylogeny (*Figure 3B*) to reconstruct ancestral gene expression states for each gene (ancestral transcriptome reconstruction). To explore the similarity of extant and reconstructed transcriptomes we used Fuzzy C-Means (FCM) clustering, a 'soft' clustering method that allows each sample to have membership in multiple clusters and assigns samples to clusters based on their degree of cluster membership. FCM with two to four clusters (K=2–4) had a clear biological interpretation (*Figure 4*): (1) K=2 clustered eutherians and non-eutherians; (2) K=3 clustered most therians with non-invasive (epitheliochorial) placentas, eutherians with invasive (endotheliochorial or hemochorial) placentas, platypus and sauropsids (i.e. viviparous and oviparous lizards, and birds); (3) K=4 clustered eutherians with invasive placentas, eutherians with epitheliochorial placentas, opossum/wallaby, and sauropsids. A notable exception with K=3–4 is the cluster membership of dunnart, which is discussed in greater detail below. FCM clusters with K=5 and K=6 were similar to K=4, but divided Eutherians with hemochorial placentas into two clusters, and clustered dog, dunnart, and the viviparous skink *Chalcides ocellatus* (*Figure 4*); Beyond K=6 clusters had no clear biological interpretation.

Ancestral transcriptome reconstructions generally clustered with extant species having similar parity modes and degrees of placenta invasiveness (*Figure 4*). For example, FCM with K=2 grouped extant eutherians and their ancestral lineages as well as extant non-eutherians and their ancestral lineages. Similarly, FCM with K=4 grouped extant eutherians with invasive placentas and their ancestral lineages with extant therians with non-invasive placentas and their ancestral lineages. FCM with K=3–5 clustered the ancestral eutherian (AncEutheria) transcriptome with extant species that have invasive hemochorial placentas and clustered the ancestral therian (AncTheria) and mammalian (AncMammalia) transcriptomes with extant mammals that have non-invasive epitheliochorial placentas. These data suggest that FCM clustering of extant and ancestral reconstructions of endometrial transcriptomes can predict ancestral placenta invasiveness, implying that an epitheliochorial placenta evolved early in the development of mammalian pregnancy and that a hemochorial placenta is ancestral for eutherians.

While FCM clustering generally groups extant and ancestral transcriptomes by phylogenetic relatedness and degree of placental invasiveness, a notable exception is the marsupial fat-tailed dunnart (*Sminthopsis crassicaudata*). FCM clusters with K=2–5 grouped dunnart with non-therians while FCM K=6 clustered dog, dunnart, and the skink, *C. ocellatus* (*Figure 4*). FCM cluster membership coefficients of all three species with K=2–5 were mixed, with significant membership across clusters. In

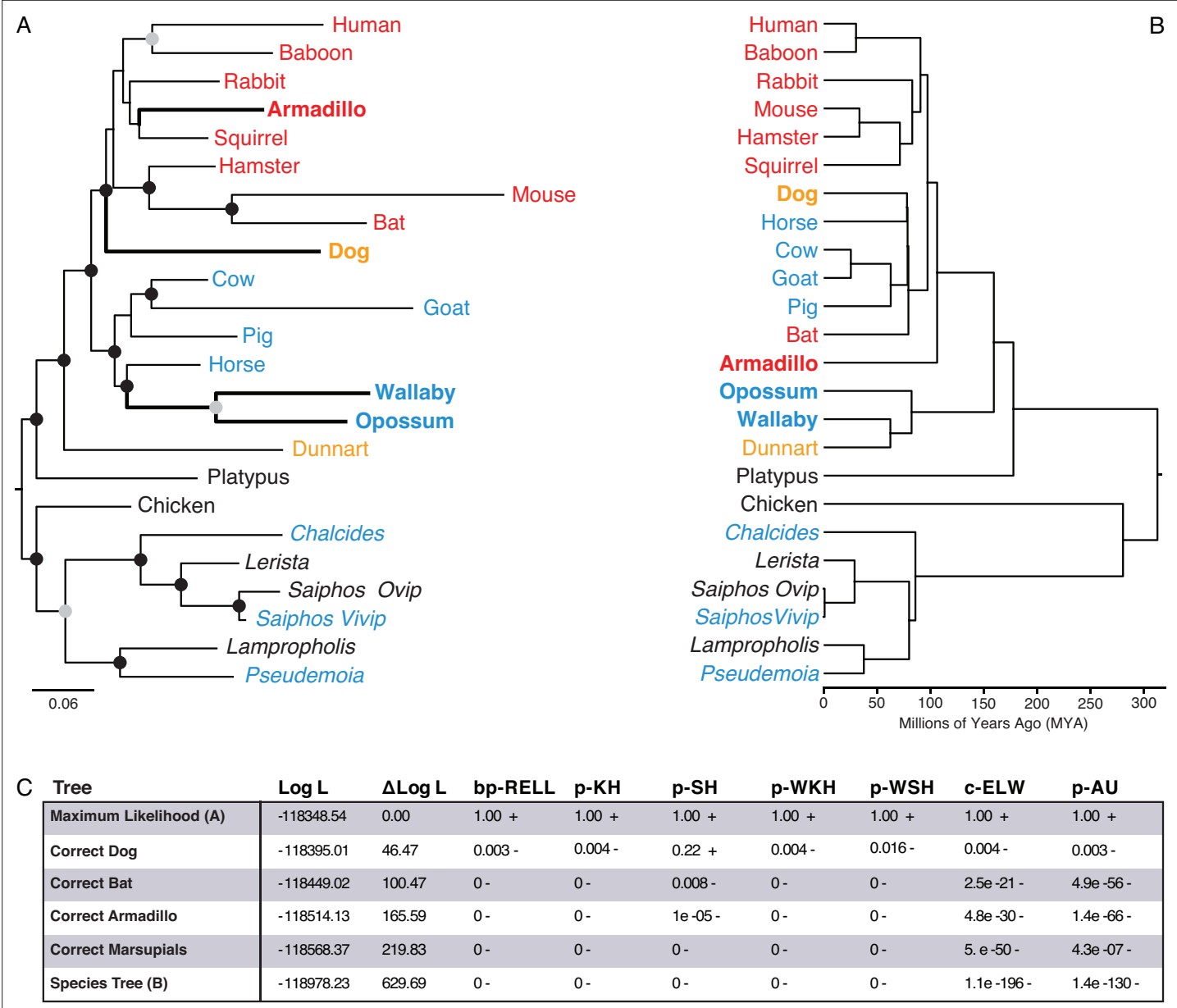

**Figure 3.** Maximum-likelihood (ML) phylogeny of binary encoded endometrial transcriptome data. (**A**) ML phylogeny of binary encoded endometrial transcriptome data inferred by IQ-TREE under the GTR2 +FO + R3 model. Highly supported branch splits, i.e., those with SH-aLRT ≥80%, LBoot ≥90%, aBayes ≥0.90, UFboot ≥95%, StdBoot support ≥80%, and parametric aLRT ≥0.95, are shown with black circles. Branch splits that are highly-supported by at least 4 of 6 methods are shown with gray circles. Particularly discordant phylogenetic relationships are shown in bold. Oviparous species are in black, viviparous species with epitheliochorial placentas in blue, endotheliochorial placentas in orange, and hemochorial placentas in red. Lizard species are shown with genus names in italics. (**B**) Phylogenetic relationships of species in (**A**). Discordant phylogenetic relationships compared to the ML tree are shown in bold. (**C**) Tree topology tests comparing the maximum likelihood tree (GTR2 +FO + H4) shown in panel A to alternate trees that correct the phylogenetic placement of specific species or the species tree shown in panel B. Delta Log L: log L difference from the maximal Log L in the set (**A**). bp-RELL: bootstrap proportion using RELL method. p-KH: p-value of one sided Kishino-Hasegawa test. p-SH: p-value of Shimodaira-Hasegawa test. p-WKH: p-value of weighted KH test. p-WSH: p-value of weighted SH test. c-ELW: Expected Likelihood Weight. p-AU: p-value of approximately unbiased (AU) test. +indicates tree is within the 95% confidence sets, – significant exclusion of tree from the 95% confidence sets. All tests performed 100,000 resamplings using the RELL method.

The online version of this article includes the following figure supplement(s) for figure 3:

**Figure supplement 1.** ML phylogeny of binary encoded endometrial transcriptome data inferred by IQ-TREE under the best-fitting GTR2 +FO + R3 model.

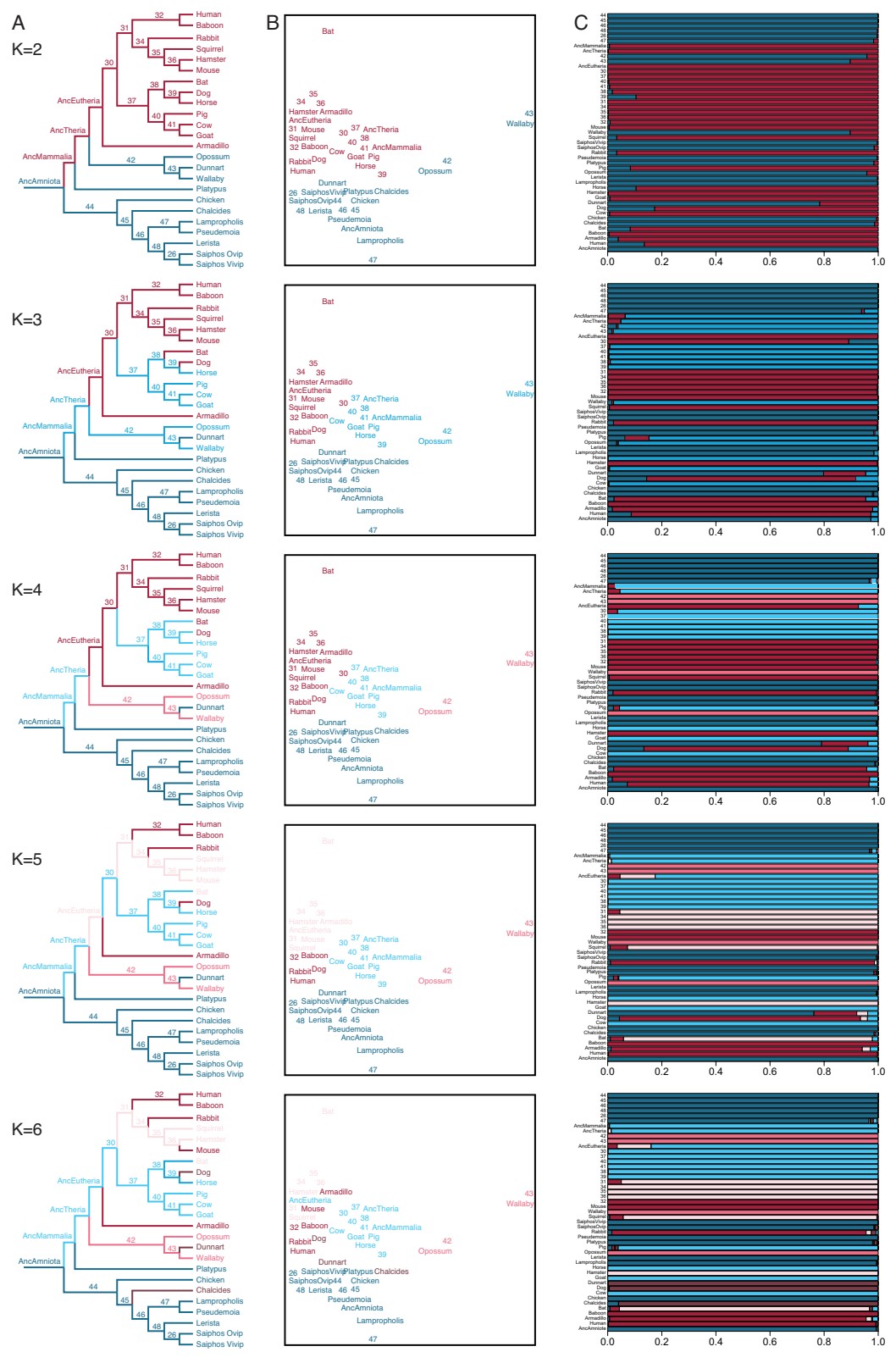

**Figure 4.** Fuzzy C-Means clustering of extant and ancestral transcriptomes. (**A**) Species phylogeny with extant species and ancestral lineages colored according to Fuzzy C-Means cluster membership (K=2–6) shown in (**B**). Note that ancestral nodes are numbered. Lizard species are shown with genus names in italics. (**B**) Fuzzy C-Means clustering of extant and ancestral transcriptomes (K=2–6). Extant species and ancestral lineages colored

*Figure 4 continued on next page*

*Figure 4 continued*

according to maximum degree of cluster membership shown in (**C**). Lizard species are shown with genus names in italics. (**C**) Fuzzy C-Means cluster membership of extant and ancestral transcriptomes (K=2–6). Degree of cluster membership is shown as a 100% stacked bar and colored according to proportion of membership in each cluster. Lizard species are shown with genus names.

The online version of this article includes the following figure supplement(s) for figure 4:

**Figure supplement 1.** Exploratory clustering of extant and ancestral binary encoded transcriptomes.

**Figure supplement 2.** Exploratory analyses of Fuzzy C-Means cluster number.

contrast, dog, dunnart, and *C. ocellatus* formed a distinct cluster from all other species at K=6 with nearly 100% FCM cluster membership in group 6 (*Figure 4*). Remarkably, both dog and dunnart have endotheliochorial placentas whereas the *Chalcides* maternal-fetal interface has been described as either endotheliochorial or epitheliochorial with extensive vascularization and interdigitating folds of hypertrophied uterine and chorioallantoic tissue (*Blackburn, 1993*; *Blackburn and Callard, 1997*; *Corso et al., 2000*). These data suggest that species with endotheliochorial placentas have a gene expression profile that is intermediate between epitheliochorial or hemochorial placentas, yet is also distinct, and that gene expression at the *Chalcides* maternal-fetal interface are converging on a therian-like endotheliochorial pattern.

## Convergent loss of *RORA* in species with epitheliochorial placentas

Among the genes with discordant species- and gene-expression phylogenies is *RAR-related orphan receptor alpha* (*RORA*), an orphan nuclear hormone receptor that regulates the development and function of type 2 and type 3 innate lymphoid cells (ILC2 and ILC3), which negatively regulates the immune system in local microenvironments especially during inflammation (*Haim-Vilmovsky et al., 2019*), the establishment of tolerance to intestinal microbiota (*Lyu et al., 2022*), and the regulation of

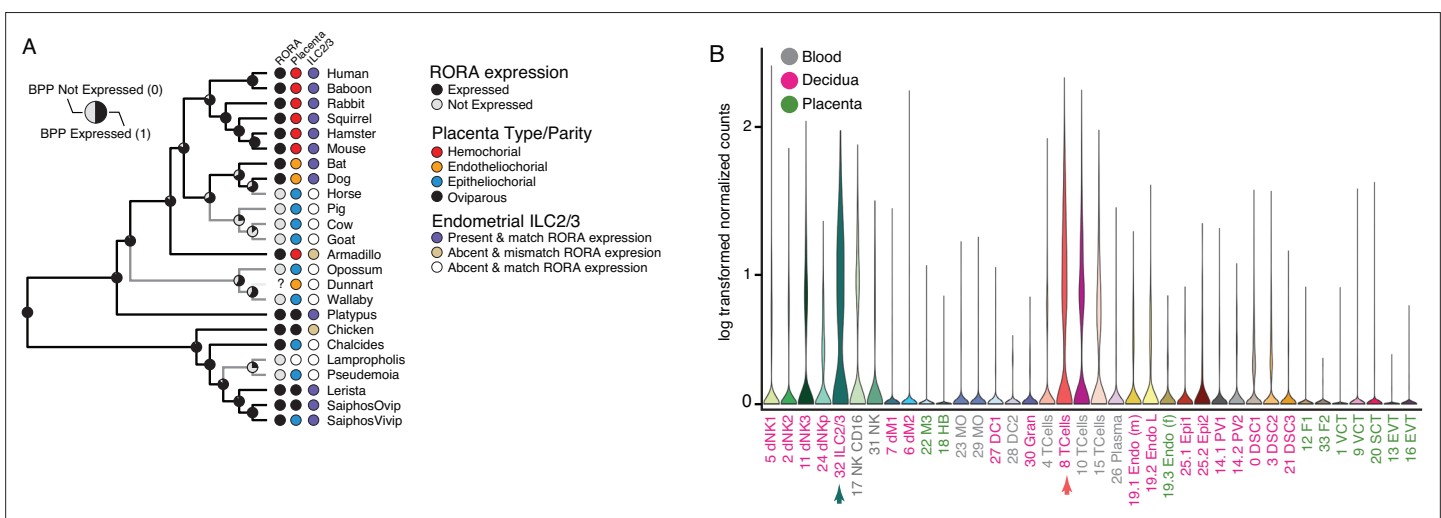

**Figure 5.** Convergent loss of *RORA* expression in species with epitheliochorial placentas. (**A**) Ancestral reconstruction of *RORA* expression. Pie charts at each internal node indicate the Bayesian Posterior Probability (BPP) that *RORA* is expressed (state 1, black) or not expressed (state 0, gray) at that node. Extant species are colored according to *RORA* expression (TPM ≥2 = expressed, black; TPM <2 = not expressed, gray); note that the RORA is not annotated in the dunnart genome and is therefore indicated as '?'. Degree of placental invasiveness is shown for extant species: Hemochorial (red), endotheliochorial (orange), epitheliochorial (blue), oviparous (black). Inference of an endometrial population of ILC2/3 cells from deconvolution of bulk RNA-Seq datasets from each species is shown for extant species: Species with ILC2/3 cells and *RORA* expression (blue dots), species without ILC2/3 cells and *RORA* expression (yellow dots), and species without ILC2/3 cells and without *RORA* expression (white). (**B**) *RORA* expression in single-cell RNA-Seq data from cell-types at the first trimester human maternal-fetal interface. Expression data is shown as a violin plot of log transformed counts for each cell-type. Cell-types are color coded by their location at the maternal-fetal interface: Blood (grey), decidua (pink), placenta (embryonic chorion, green).

The online version of this article includes the following source data for figure 5:

**Source data 1.** Nexus file of the binary encoded endometrial transcriptome dataset with gene names and species phylogeny.

inflammatory responses and vascular remodeling during placentation and pregnancy (*Balmas et al., 2018*; *Mendes et al., 2020*; *Miller et al., 2018*). Remarkably, *RORA* independently lost endometrial expression at least four times, each loss coincident with the independent evolution of non-invasive epitheliochorial placentas (*Figure 5A*). The restricted expression of *RORA* to immune cells in the human first trimester decidua (*Vento-Tormo et al., 2018*; *Figure 4B*) suggests that the repeated loss of *RORA* expression reflects changes in the composition of immune cells at the maternal-fetal interface in species with epitheliochorial placentas. To explore this possibility, we used CIBERSORT to deconvolve endometrial bulk RNA-Seq datasets from each species into cell-type abundance estimates with a signature gene expression matrix composed of cell-types from the human first trimester maternal-fetal interface. We found that CIBERSORT inferred that nearly all species that lacked *RORA* expression also lacked ILC2/3 cells at the maternal-fetal interface with the exception of armadillo and chicken which had *RORA* expression but were inferred to lack endometrial ILC2/3 cells, and the viviparous *Saiphos* population which has an epitheliochorial placenta and *RORA* expression and was inferred to have a population of ILC2/3 cells. Thus, we conclude that there is convergent loss of *RORA* in species with epitheliochorial placentas which in many species correlates with the convergent loss of ILC2/3 cells at the maternal-fetal interface.

## Discussion

One of the central of goals of DevoEvo is a mechanistic explanation for the origin and evolution of evolutionary novelties (*Amundson, 2005*; *Brigandt and Love, 2010*; *Lynch, 2022*; *Wagner, 2001*; *Wagner, 2000*; *Wagner and Larsson, 2003*). A major challenge for reconstructing the origins of evolutionary novelties, however, is a lack of transitional forms among living species. For example, while feathers and the turtle's shell are excellent examples of morphological novelties, there are no living species with transitional forms between scale and feather (*Chen et al., 2015*) or with protoshells (*Lyson and Bever, 2020*). Despite the lack of transitional forms among extant taxa, an abundance of fossil data has reconstructed the major steps in the evolution of these structures that when combined with molecular studies of their development provides a rich explanation for their developmental evolution. Unfortunately, many features of soft tissues leave little to no trace in the fossil record, thus the steps in their evolution have remained elusive. Here, we used ancestral transcriptome reconstruction to trace gene expression changes during the origins of mammalian pregnancy, in which extant species preserve intermediate stages in the evolution of both pregnancy and a diversity of placenta types and reconstruct the evolution of placental invasiveness of ancestral species.

### Binary encoding uncovers hidden biological signal

While transforming gene expression count data into binary categories likely loses information about evolutionarily relevant variation in expression levels, it may not be possible to infer meaningful gene expression levels from bulk RNA-Seq. For example rather than evolutionary differences between individuals or species, variation in transcript abundance between samples can result from various sources of experimental noise such as technical variation in library preparation, sequencing, or batch effects (*Gilad and Mizrahi-Man, 2015*; *Tung et al., 2017*), variation in cell-type composition of a tissue (*Price et al., 2022*), sampling different timepoints in development or only a few individuals that do not capture the variance properties of gene expression levels within a population or species (*Pal et al., 2020*; *Thompson et al., 2020*). Thus, by transforming gene expression data into not/expressed states we may reduce the potential for these and other biases to influence our ancestral transcriptome reconstructions, revealing biological signal. Our binary encoded endometrial gene expression dataset, for example, cluster species with similar placenta types and parity mode (*Figure 2B*) indicating binary encoding preserves functional signal in gene expression data that is otherwise masked by variation in gene expression levels (*Figure 2A*).

### Phylogenetic analyses of endometrial transcriptomes

Previous studies of molecular phylogenies have shown that gene-species tree discordance can result from convergent or parallel amino acid changes in lineages that independently evolved morphological traits. For example, parallel amino acid changes in *prestin* (*SLC26A5*), a motor protein expressed in outer hair cells in the cochlea that essential for hearing, have occurred in different lineages of

echolocating mammals, including multiple lineages bats and whales, leading a strongly monophyletic clades in gene trees that do not reflect taxonomic relationships (*Li et al., 2008*; *Li et al., 2010*; *Liu et al., 2010*; *Teeling, 2009*). Gene-species tree discordance is also associated coadaptation of amino acid substitutions in Na⁺K⁺–ATPase of frogs that prey on toxic toads (*Mohammadi et al., 2021*), the evolution of $C_4$ *phosphoenolpyruvate carboxykinases* (*PCK*) in grasses (*Christin et al., 2009*), and between snake and agamid lizard mitochondrial genomes (*Castoe et al., 2009*). These examples highlight how phylogenetic discordance can be a signal of convergent evolution, rather than just a sign of biased phylogenetic inference that can arise from processes like long branch attraction (*Bergsten, 2005*; *Felsenstein, 1978*), incomplete lineage sorting and introgression (*Guerrero and Hahn, 2018*; *Hibbins et al., 2020*; *Maddison and Knowles, 2006*), biased character state frequencies and gene conversion (*Figuet et al., 2014*; *Kostka et al., 2012*; *Lartillot, 2013*; *Romiguier et al., 2013*), and heterotachy (*Kolaczkowski and Thornton, 2008*; *Philippe et al., 2005*), among others.

Similar to the signals of convergence from gene-species tree discordance, our transcriptomic data showed significant phylogenetic support for uterine transcriptomes grouping by parity mode and degree of placental invasiveness rather than phylogenetic relationships. This transcriptome-species tree discordance is particularly striking for opossum and wallaby, which are deeply nested within eutherians with epitheliochorial placentas based on uterine transcriptome data, because the eutherian and marsupial placenta is derived from different extra-embryonic tissues – the chorion and allantois in eutherians and the yolk-sac in marsupials. Thus, there must be significant convergence in *endometrial* gene expression profiles between eutherians with chorioallantois-derived epitheliochorial placentas and marsupials with yolk-sac epitheliochorial placentas because our transcriptome phylogeny is based on gene expression in the endometrium during pregnancy rather than gene expression in the placenta. This convergence in gene expression between species with non-invasive epitheliochorial placentas may be related to the expression of genes that limit the ability of the trophoblast to invade into maternal tissues (see below).

## Reconstruction of ancestral (endometrial) transcriptomes

Previous studies have explored transcriptome evolution with the goal of developing methods of ancestral transcriptome inference (*Price et al., 2022*), characterizing the general tempo and mode of gene expression evolution (*Bauernfeind et al., 2021*), implicating transcriptome evolution in the origin and evolution of morphological traits (*Church et al., 2021*; *Mika et al., 2021b*; *Lynch et al., 2015*; *Marinić et al., 2021*; *Munro et al., 2021*), and identifying specific genes with derived expression levels (*Brawand et al., 2011*; *Necsulea et al., 2014*), usually in the context of characterizing genes whose expression levels have changed because of the action of positive selection, that is, directional selection on transcript abundance (*Gu, 2004*; *Price et al., 2022*; *Yang et al., 2020*). In contrast, only a few studies have treated the expression of individual genes as a discrete character, transforming quantitative gene expression levels into binary not/expressed states, and used ancestral state reconstruction to determine the expression state of all genes in an ancestral transcriptome rather than changes in the expression levels of specific genes (*Mika et al., 2021b*; *Kin et al., 2015*; *Lynch et al., 2015*; *Marinić et al., 2021*). Our ancestral transcriptome reconstructions of gene expression state indicate that an epitheliochorial-like placenta evolved early in the mammalian stem-lineage, before the loss of the egg-shell in Therians, and that the ancestor of Eutherians had a hemochorial placenta, which resolves a longstanding debate about the nature of ancestral mammalian placentas. These data suggest that ancestral transcriptome reconstruction can be used to infer the function of ancestral cell, tissue, and organ systems which leave little to no trace in the fossil record even if soft tissues might by chance fossilize.

## Convergent loss of *RORA* in species with epitheliochorial placentas

Our observation that *RORA* expression has been lost multiple times in species with non-invasive epitheliochorial placentas, which is coincident with the inferred absence of ILC2/ILC3 cells from the endometrium during at least one stage in pregnancy. This result suggests a mechanistic connection between absence of decidual *RORA* expression, the absence of decidual ILC2/ILC3 cells, and the loss of placental invasion. Indeed, the major decidual ILC produce signaling factors such as GM-CSF, XCL1, MIP1α, and MIP1β, whose receptors are expressed by EVT likely regulate placental invasion (*Huhn et al., 2020*). ILC2 also contributes to the maintenance of a type-2 anti-inflammatory immune

environment in the uterus during pregnancy (*Balmas et al., 2018*), which may be particularly important in species in which the endotheliochorial or haemochorial placenta invades maternal tissues. ILC2s, for example, are increased in the decidua basalis of women with spontaneous preterm labor compared to those who delivered preterm without labor (*Mendes et al., 2021*; *Xu et al., 2018*), suggesting that ILCs may participate in the chronic inflammatory process that occurs during pregnancy. Thus, loss of placental invasion may be associated with a reduced need to limit local inflammation by ILCs, leading to an evolutionary loss of ILCs in the endometrium during pregnancy. However, while our data are consistent this role for decidual ILC2 and its association with placental invasion, more detailed studies specifically aimed at characterizing cell-type composition differences across species are necessary to determine if there are correlations between cell-types in the endometrium during pregnancy and placenta type.

## Ideas and speculation: Maternal control of placental invasion

Our observation that endometrial gene expression patterns are correlated with degree of placental invasiveness might seem surprising, however, rather than acting as a passive substrate into which the trophoblast invades, the endometrium directly controls trophoblast invasion (*Cui et al., 2012*; *Graham and Lala, 1991*). For example, the trophoblast of mammals with hemochorial placentas, such as humans and rodents, is only permissive to invasion when the 'window of implantation' is opened by the endometrium. Similarly, while the trophoblast of mammals with non-invasive endotheliochorial and epitheliochorial placentas, such as cats, dogs, horses, cows, pigs, and sheep cannot invade into the endometrium, they can invade into ectopic sites (reviewed in *Corpa, 2006*). The trophoblasts of guinea pig (*Loeb, 1914*), mouse (*Billington, 1965*), rat (*Jollie, 1961*), and pig (*Samuel and Perry, 1972*) also invade ectopic sites in experimentally induced ectopic pregnancy. While there is no similar ectopic pregnancy data for marsupials, some marsupial lineages, including dunnart, have evolved invasive placentation. In contrast, there is no invasion of maternal tissues during ectopic pregnancy in viviparous reptiles with epitheliochorial placentation such as *Pseudemoia entrecasteauxii* (*Griffith et al., 2013*). Thus, the invasive ability of trophoblasts most likely either evolved in the stem-lineage of therian mammals or multiple times, including in the stem-lineage of eutherians and some lineages of marsupials. Regardless, ancestral maternal control of placental invasion likely allowed us to infer ancestral placental invasiveness from ancestral endometrial transcripomes.

## Ideas and speculation: Evolution of placental invasion and cancer metastasis

Numerous authors have noted the similarity between placental invasion and cancer metastasis (*Costanzo et al., 2018*; *Ferretti et al., 2007*; *Kozlov, 2022*; *Lala et al., 2021*; *Manzo, 2019*; *Murray and Lessey, 2008*; *Perry et al., 2009*; *Piechowski, 2019*), which was first proposed in 1902 by Scottish embryologist John Beard (1858–1924) who hypothesized that ectopic trophoblasts gave rise cancer (*Gurchot, 1975*; *Ross, 2015*). While Beard's hypothesis is incorrect, at least for cancers not derived from the placenta such as choriocarcinoma, there are numerous mechanistic similarities between implantation, placental invasion, and tumor progression to malignancy (*Nordor et al., 2017*; *Wagner et al., 2022*). These data suggest that the evolution of maternal mechanisms that prevent endometrial invasion through expression gain and loss of genes that restrain and promote, respectively, trophoblast invasion, may be related to resistance to metastasis in eutherian lineages with non-invasive placentas (*Boddy et al., 2020*; *Afzal et al., 2019*; *Wagner et al., 2020*). For example, pleiotropy can lead to correlated patterns of gene expression between the transcriptomes of different tissue and organ systems (*Liang et al., 2018*). Thus, the evolution of a gene regulatory module that restricts placental invasion into the endometrium can be coopted to restrict implantation and spread of cancer cells into metastatic locations.

## Caveats and limitations

A limitation of this study not directly addressed thus far is that we have only sampled a small number of species. For example, we lack pregnant endometrial samples from most mammals, particularly those with endotheliochorial placentas, as well as a diversity of oviparous and viviparous squamates there are at least 115 origins of viviparity in squamates (*Blackburn, 2015*; *Blackburn and Brandley, 2015*; *Blackburn and Starck, 2015*). Thus, our inferences from phylogenetic, ancestral reconstruction,

and clustering analyses may be biased by small sample sizes and non-random sampling. We also assume that models of evolution designed for phylogenetic inference and ancestral reconstruction of morphological and molecular data are appropriate for gene expression data or binary encoded gene expression data, which may affect our results; for example, we have not directly accommodated incomplete lineage sorting which can mislead phylogenetic inference (*Guerrero and Hahn, 2018*; *Hibbins et al., 2020*). Similarly, while Fuzzy C-Means clustering is conceptually similar to topic ('grade of membership') models used in population genetics, its underlying assumptions may be violated for gene expression and binary encoded gene expression data. More detailed studies are necessary to determine if our results are robust to potential sources of error such as model mis-specification, small sample sizes, and non-random taxon sampling, as well as incomplete lineage sorting, the latter of which we were unable to directly test.

## Conclusions

Previous critiques of statistical methods to infer ancestral states, particularly in the context of parity mode evolution in squamates, have suggested that ancestral state reconstructions of morphological characters must be supported by additional kinds of biological support such as anatomical, physiological, and ecological evidence, to be persuasive (*Griffith et al., 2015*). Here we explored the evolution of parity mode and placental invasiveness in amniotes utilizing comparative gene expression data. While our study also relies on statistical methods to infer ancestral (gene expression) states, this approach is orthogonal to traditional methods that infer ancestral states from morphological characters among extant species. Indeed, gene expression ultimately underlies the development, evolution, and function of anatomical systems. Thus, by reconstructing the evolution of entire transcriptomes we may be able to infer function of ancestral cell, tissue, and organ systems. Our results resolve several evolutionary transformations during the origins of pregnancy, including the early evolution of an epitheliochorial-like placenta in the mammalian stem-lineage, a hemochorial placenta in the ancestor of eutherians, multiple reversions to non-invasive epitheliochorial placentas within some eutherian lineages, convergent evolution of gene expression profiles among species with different ontogenetic origins of epitheliochorial placentas, and maternal control of placental invasiveness.

## Materials and methods

### Endometrial gene expression profiling

We previously published a dataset of uterine endometrial transcriptomes, which is also used in this study. Interested readers are referred to *Mika et al., 2021b* for specific details. Briefly, we searched the NCBI BioSample, Short Read Archive (SRA), and Gene Expression Omnibus (GEO) databases using the search terms 'uterus', 'endometrium', 'decidua', 'oviduct', and 'shell gland'. These anatomical terms refer to the glandular portion of the female reproductive tract, which is specialized for maternal-fetal interactions or shell formation. We then manually curated transcriptomes and excluded those that did not indicate whether tissue samples were from pregnant or gravid tissues and datasets composed of pooled tissues. Gene expression data were analyzed with Kallisto (*Bray et al., 2016*) version 0.42.4 to pseudo-align the raw RNA-Seq reads to reference transcriptomes and to generate transcript abundance estimates (see *Figure 2—source data 1* for accession numbers and reference genome assemblies); Kallisto was run using default parameters, bias correction, and 100 bootstrap replicates.

### Gene expression phylogeny and ancestral transcriptome reconstruction

We used the binary encoded endometrial transcriptome dataset for phylogenetic analyses and to reconstruct ancestral gene expression states. Gene expression phylogenies were inferred with IQ-TREE2 (*Nguyen et al., 2015*) using the best-fitting model of character evolution determined by ModelFinder (*Kalyaanamoorthy et al., 2017*). The best fitting model was inferred to be the General Time Reversible model for binary data (GTR2), with character state frequencies optimized by maximum-likelihood (FO), and a FreeRate model of among site rate heterogeneity with three categories (R3) (*Soubrier et al., 2012*). The rate at which characters evolve may vary over time, with the same character evolving rapidly or slowly in different lineages. This phenomenon, known as heterotachy, can bias phylogenetic trees using models of evolution that assume the rates of character evolution

are constant such as the GTR +FO + R3 model. Therefore we compared the GTR2 +FO + R3 model to the General Heterogeneous evolution On a Single Topology (GHOST) model; The GHOST model accommodates heterotachy by combining features of mixed substitution rate models (*Foster, 2004*; *Lartillot, 2013*; *Pagel and Meade, 2004*), whereby each class of characters has its own substitution rate matrix, and mixed branch length models (*Kolaczkowski and Thornton, 2008*; *Pagel and Meade, 2004*), whereby each class of characters has its own set of branch lengths.

We found the best-fit GHOST model was GTR2 +FO*H4 (AICc = 215681.71), which accommodated rate heterotachy with 4 categories, described the binary encoded gene expression dataset better than the standard GTR2 +FO + R3 model (AICc = 220930.65) indicating that there is extensive variation in the rate of character evolution over time. However, while there was a significant likelihood difference between GTR2 +FO + R3 and GTR2 +FO*H4 (AICc difference = 5248.94), as well as other heterotachy with models with less than or more than 4 categories, the topology of the trees was the same. Thus, while accommodating heterotachy improves estimation of parameters of the substitution model it had no affect on the tree topology and we therefore use the computationally simpler GTR2 +FO + R3 model for downstream analyses; This is particularly important for non-parametric topology tests and ancestral state reconstructions (discussed below), which as currently implemented in IQ-TREE2 cannot accommodate heterotachy models with unlinked model parameters.

Ancestral gene expression states for each gene were inferred using the empirical Bayesian method implemented in IQ-TREE2, the GTR2 +FO + R3 model of character evolution, and the species phylogeny as a constraint tree (*Figure 3B*). Branch support was assessed using the standard (StdBoot) and ultrafast (UFBoot) bootstraps, which assess the effects of sampling bias on branch support (*Hoang et al., 2018*; *Minh et al., 2020*). We also used several single branch tests, including the SH-like aLRT and the parametric aLRT (*Anisimova et al., 2006*; *Guindon et al., 2010*) aBayes (*Anisimova et al., 2011*) and the local (LBoot) bootstrap tests (*Minh et al., 2020*) single branch tests assess whether a branch provides a significant likelihood improvement compared to a null hypothesis that collapses the branch to a polytomy but leaves the rest of the tree topology unaltered. We considered a clade to be highly-supported if its StdBoot support ≥80%, UFboot ≥95%, SH-aLRT ≥80%, aBayes ≥0.90, parametric aLRT ≥0.95, and LBoot ≥90% (*Anisimova et al., 2011*).

The bootstrap and single branch tests assess the robustness of individual branch bipartitions and cannot directly compare complex alternate tree topologies. Therefore we used non-parametric topology tests to directly compare the inferred ML tree to alternative trees with the correct phylogenetic placement of platypus, armadillo, dog, marsupials (opossum and wallaby), and bat, as well as the correct species phylogeny (*Figure 2B*); tests included the BP-RELL, KH-test (*Kishino et al., 1990*; *Kishino and Hasegawa, 1989*), SH-test (*Anisimova et al., 2011*; *Guindon et al., 2010*), c-ELW (*Strimmer and Rambaut, 2002*), weighted KH- and SH-tests, and the AU-test (*Shimodaira and Goldman, 2002*). We note that the KH-test compares two a priori defined trees rather than the ML and alternative trees (*Goldman et al., 2000*) and does not correct for multiple hypothesis tests, it is included solely for comparison to other methods and previous studies. The SH-test can be used to compare the ML tree to multiple alternative trees selected a priori (i.e. is dataset independent) and corrects for multiple hypothesis tests, but is too conservative when many trees are tested. The AU-test, in contrast, resolves the conservative nature of the SH-test and thus is the preferred test. All tests performed 100,000 resamplings using the RELL method.

## Clustering methods

We evaluated multiple methods to summarize and visualize the binary encoded extant and ancestral reconstructed transcriptomes, including: (1) Logistic Principal Component Analysis (LPCA), a version of principal component analysis for dimensionality reduction of binary data (https://cran.r-project.org/web/packages/logisticPCA/vignettes/logisticPCA.html); (2) classical Multi-Dimensional Scaling (MDS); (3) Uniform Manifold Approximation and Projection (UMAP); (4) tSNE; and (5) Fuzzy C-Means (FCM) clustering. All clustering analyses were conducted in R after removing columns (genes) with missing data (coded as?/NA) or that were invariant (all 0 or all 1). LPCA was performed using the LogisticPCA R package (*Landgraf and Lee, 2015*), which implements three methods: exponential family PCA applied to Bernoulli data, logisitic PCA, and the convex relaxation of logistic PCA. For each of the methods, we fit the parameters assuming two-dimensional representation, returning four principal components (ks = 4), and selecting the best m value to approximate the saturated model for cross

validation. MDS was performed using the vegan R package (*Oksanen et al., 2008*) with four reduced dimensions. UMAP was performed using the umap R package. tSNE was performed using the Rtsne R package.

To explore the data in greater detail we focused on FCM clustering because the results were qualitatively similar to the other methods, and it has several desirable properties including providing a statistically sound way to identify clusters rather than an ad hoc approach that might be applied to the other methods. FCM also allows each sample to have membership in multiple clusters and is conceptually similar to topic ('grade of membership') models used in population genetics to visualize private and shared genetic structure across populations. FCM membership coefficients can thereby account for multiple sources of similarity including noise, phylogenetic signal, and convergence of gene expression. FCM was performed in R using the R package, using Manhattan distances (cluster membership was not altered by using other distance metrics), and an estimated fuzzifier (m=1.034978). FCM clustering requires a priori knowledge of the number of clusters (K) to include, therefore we evaluated FCM with K=2–9 following the suggestions given in https://www.r-bloggers.com/2019/01/10-tips-for-choosing-the-optimal-number-of-clusters/. First, we used the "elbow" method, in which the sum of squares of each cluster number is calculated and graphed and the optimal number of clusters estimated by a change of slope from steep to shallow (the elbow). We also assessed the optimal number of clusters using the clustree R package, which assess the optimal number of clusters by considering how samples change groupings as the number of clusters increases ; clustree is useful for estimating which clusters are distinct and which are unstable but cannot determine the optimal number of clusters (K).

## CIBERSORT analyses

We inferred the proportion of different cell-types in the endometrium during pregnancy across our comparative gene expression datasets using CIBERSORT (*Newman et al., 2019*), which takes as input a file of gene expression levels from a mixed cell population and a gene expression signature file with expression levels of marker genes in specific cell-types. CIBERSORT was run on the bulk RNA-Seq data from each species using a signature gene expression file based on the Vento-Tormo et al. scRNA-Seq dataset. The signature gene expression file included gene expression data (TPM-like) for genes with an expression level greater than or equal to the expression threshold that also have at least fivefold higher expression levels in a particular cell compared to all other cells (*Jain and Tuteja, 2021*).

## Acknowledgements

The authors thank MB Thompson (University of Sydney) for constructive comments on this manuscript. This study was supported by a grant from the March of Dimes (March of Dimes Chicago-Northwestern-Duke Prematurity Research Center) and a Burroughs Welcome Fund Preterm Birth Initiative grant (1013760) to principal investigator VJL. The funders had no role in study design, data collection and analysis, decision to publish, or preparation of the manuscript.

## Additional information

### Funding

| Funder | Grant reference number | Author |
|---|---|---|
| March of Dimes Foundation | | Vincent J Lynch |
| Burroughs Wellcome Fund | 1013760 | Vincent J Lynch |

The funders had no role in study design, data collection and interpretation, or the decision to submit the work for publication.

### Author contributions

Katelyn Mika, Conceptualization, Data curation, Formal analysis, Investigation, Writing – original draft, Writing – review and editing; Camilla M Whittington, Bronwyn M McAllan, Data curation, Resources,

Writing – original draft, Writing – review and editing; Vincent J Lynch, Conceptualization, Data curation, Formal analysis, Funding acquisition, Investigation, Methodology, Project administration, Supervision, Visualization, Writing – original draft, Writing – review and editing

### Author ORCIDs
Katelyn Mika https://orcid.org/0000-0002-2170-9364
Camilla M Whittington https://orcid.org/0000-0001-5765-9699
Vincent J Lynch https://orcid.org/0000-0001-5311-3824

### Decision letter and Author response
Decision letter https://doi.org/10.7554/eLife.74297.sa1
Author response https://doi.org/10.7554/eLife.74297.sa2

## Additional files

### Supplementary files
• Figure 2—source data 2. file 2. Binary encoded endometrial transcriptome dataset.
• MDAR checklist

### Data availability
All data generated or analysed during this study are included in the manuscript and supporting file; Source Data files have been provided for Figures 2, 3 and 5.

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
