## [Editor Report]

Mika and colleagues reconstruct the evolution of uterine endometrial transcriptomes during pregnancy from 23 diverse species of mammals that differ with respect to their degree of placental invasiveness. Through this analysis, the authors infer that the eutherian mammal ancestor had an invasive mode of placentation and that the degree of invasiveness of placentation is reflected on uterine endometrial gene expression during pregnancy. Thus, phylogenetic analysis of gene expression profiles of different mammals groups them on the basis of the degree of placental invasiveness, a quite striking finding.

---

## [Decision Letter]

**Decision letter after peer review:**

Thank you for submitting your article "Gene expression phylogenies and ancestral transcriptome reconstruction resolves major transitions in the origins of pregnancy" for consideration by *eLife*. Your article has been reviewed by 3 peer reviewers, one of whom is a member of our Board of Reviewing Editors, and the evaluation has been overseen by George Perry as the Senior Editor. The following individual involved in review of your submission has agreed to reveal their identity: Adam Stevens (Reviewer #3).

Essential revisions:

1) Explore which transcripts (or what functional categories) drive the convergence signal.

2) Revise the statement concerning the maternal as opposed to the fetal control of placental invasiveness. We understand that the evidence supports this idea but without direct comparison to fetal data it is difficult to accept as stated.

3) Does ancestral transcriptome reconstruction take into account gene-level processes, such as introgression and lineage sorting of ancestral polymorphisms? It is will established that gene phylogenies often differ from species phylogenies, so it is important that the authors elaborate on this (at least in the caveats section). The recent work by the Hahn group on hemiplasy (e.g., https://www.pnas.org/content/115/50/12787 and https://elifesciences.org/articles/63753) is very relevant to this issue.

*Reviewer #1 (Recommendations for the authors):*

– I think the statement that "placental invasiveness is regulated by gene expression profiles in the maternal endometrium rather than the fetal portion of the placenta" is premature. One has to study the degree of convergence in the evolution of the fetal placental transcriptomes across the same species to make this claim.

– Can you explore which transcripts (or what functional categories) drive the convergence signal?

*Reviewer #2 (Recommendations for the authors):*

The manuscript is generally well written and easy to understand. I believe some of the conclusions could be strengthened by more explicitly explaining how the data supports these conclusions. For example, pointing out how/why the phylogenetic trees mean "there must be significant convergence in endometrial gene expression" would be helpful in interpreting the validity of this conclusion.

I also suggest coloring the species in Figure 1 using the same schema as in Figure 3. This would allow the reader to more easily see which species in Figure 1A are hemochorial, endotherliochorial, etc.

*Reviewer #3 (Recommendations for the authors):*

Can the authors expand on the issues associated with the "bulk" transcriptome of the endometrium that has been used compared to single cell RNAseq? What does this mean for the conclusions?

Statement on the bottom of page 3 concerning the maternal as opposed to fetal regulation is overstated in the opinion of this reviewer. I accept that the evidence supports this idea but without direct comparison to fetal data it is difficult to accept as stated.

Does the evolutionary modeling using the anscestral transcriptome account for features of evolution such as introgression and heterochrony? Perhaps further comment would useful.

---

## [Author Response]

Essential revisions:1) Explore which transcripts (or what functional categories) drive the convergence signal.

We did not identify specific functional categories in which genes with evidence of convergence were significantly enriched, however, we have updated the manuscript and include an example of a gene that convergently lost endometrial expression in species with epitheliochorial placentas – *RORA*. We show that *RORA* lost endometrial expression multiple times, in each case associated with the evolution of an epitheliochorial placenta. Further, RORA is specifically expressed in a subset of T-cells and innate lymphoid cells (ILC) and we show through deconvolution of our multispecies bulk RNA-Seq data that loss of RORA expression in the endometrium is correlated with loss of ILCs in species with epitheliochorial placentas. These data suggest that there are differences in the mechanisms that regulate local inflammatory signals in species with epitheliochorial placentas because ILCs play a key role in suppressing inflammation. Thus, there may be differences in the levels of uterine inflammation between species with different placenta types.

We also now include a nexus file with the binary encoded gene expression data for each species, gene names, and the species phylogeny which will allow interested readers to easily trace gene expression changes across the phylogeny with software programs like Mesquite.

2) Revise the statement concerning the maternal as opposed to the fetal control of placental invasiveness. We understand that the evidence supports this idea but without direct comparison to fetal data it is difficult to accept as stated.

We have revised statements about maternal vs fetal control of placental invasiveness throughout the manuscript and moved discussion of these topics to two new “Ideas and Speculation” sections, in which we speculate about the implications of our results for maternal control of placental invasion and the evolution of placental invasion and cancer metastasis. These new sections are titled “Ideas and Speculation: Maternal control of placental invasion” and “Ideas and Speculation: Evolution of placental invasion and cancer metastasis” and are limited to one paragraph each.

3) Does ancestral transcriptome reconstruction take into account gene-level processes, such as introgression and lineage sorting of ancestral polymorphisms? It is will established that gene phylogenies often differ from species phylogenies, so it is important that the authors elaborate on this (at least in the caveats section). The recent work by the Hahn group on hemiplasy (e.g., https://www.pnas.org/content/115/50/12787 and https://elifesciences.org/articles/63753) is very relevant to this issue.

The maximum likelihood method that we used (implemented in IQ-TREE 2) *does not* account for gene level processes such as introgression and incomplete lineage sorting but *does* account for rate variation that arises from heterotachy. Taking heterotachy first, we used the General Heterogeneous evolution On a Single Topology (GHOST) model, an edge-unlinked mixture model consisting of several site classes that each have a separate set of model parameters and edge lengths on the same tree topology and that does not require the a priori data partitioning, to explore the effects of heterotachy on our phylogenetic inference. We found that while the GHOST model fit the data better than the standard model that does not account for heterotachy the topology of the inferred tree did not change. We now include this data in the methods section in which we discuss model choice.

Unfortunately, we were not able to test for the effects of incomplete lineage sorting (ILS) on our phylogenetic inference. IQ-TREE implements two methods to infer genealogical concordance: the gene concordance factor (gCF) and the site concordance factor (sCF); gCF and sCF complement classical measures of branch support (e.g. bootstrap) by providing a full description of underlying disagreement among loci/sites and can quantify the effects of ILS in individual loci compared to species tree. While the methods to compute gCF and sCF are general, i.e., are not specific to genetic or other molecular data, we were unable to estimate either gCF or sCF using our binary encoded dataset after significant attempts to get these methods running. The source of the error is unclear.

Reviewer #1 (Recommendations for the authors):– I think the statement that "placental invasiveness is regulated by gene expression profiles in the maternal endometrium rather than the fetal portion of the placenta" is premature. One has to study the degree of convergence in the evolution of the fetal placental transcriptomes across the same species to make this claim.

See response essential revision #2 above. Also note that we have revised this statement to be:

“and *suggest* that placental invasiveness is regulated by gene expression profiles in the maternal endometrium rather than the fetal portion of the placenta.”

– Can you explore which transcripts (or what functional categories) drive the convergence signal?

See response essential revision #2 above.

Reviewer #2 (Recommendations for the authors):The manuscript is generally well written and easy to understand. I believe some of the conclusions could be strengthened by more explicitly explaining how the data supports these conclusions. For example, pointing out how/why the phylogenetic trees mean "there must be significant convergence in endometrial gene expression" would be helpful in interpreting the validity of this conclusion.

Thank you for the suggestion. We have revised this sentence to read:

“Thus, there must be significant convergence in endometrial gene expression profiles between eutherians with chorioallantois-derived epitheliochorial placentas and marsupials with yolk-sac epitheliochorial placentas because our transcriptome phylogeny is based on gene expression in the endometrium during pregnancy rather than gene expression in the placenta.”

The latter statement is intended to highlight the reason this observation is perhaps surprising.

I also suggest coloring the species in Figure 1 using the same schema as in Figure 3. This would allow the reader to more easily see which species in Figure 1A are hemochorial, endotherliochorial, etc.

This is an excellent suggestion. We have updated figure 1B to color boxes as shown in figure 3.

Reviewer #3 (Recommendations for the authors):Can the authors expand on the issues associated with the "bulk" transcriptome of the endometrium that has been used compared to single cell RNAseq? What does this mean for the conclusions?

Because we have not used single cell RNA-Seq (scRNA-Seq) we cannot directly determine the cell-types in which gene expression changes occurred. However, this limitation does not adversely affect our inferences because our major result is not dependent on knowing the cell-type in which gene expression changes happened only that gene expression changes can be inferred and reconstructed using phylogenetic methods. The question of which cell-types drive the gene expression evolution signal, however, is an interesting one.

Statement on the bottom of page 3 concerning the maternal as opposed to fetal regulation is overstated in the opinion of this reviewer. I accept that the evidence supports this idea but without direct comparison to fetal data it is difficult to accept as stated.

See response to comments above.

Does the evolutionary modeling using the anscestral transcriptome account for features of evolution such as introgression and heterochrony? Perhaps further comment would useful.

The maximum likelihood method we used can account for heterotachy but not introgression. We now note this in the “Caveats and Limitations” section. See also response to comments above.